# Survival, Development, and Fecundity of *Spodoptera frugiperda* (J.E. Smith) (Lepidoptera: Noctuidae) on Various Host Plant Species and Their Implication for Pest Management

**DOI:** 10.3390/insects14070629

**Published:** 2023-07-12

**Authors:** Ihsan Nurkomar, Dina Wahyu Trisnawati, Fajrin Fahmi, Damayanti Buchori

**Affiliations:** 1Department of Agrotechnology, Faculty of Agriculture, Universitas Muhammadiyah Yogyakarta, Jl. Brawijaya, Kasihan, Bantul, Yogyakarta 55183, Indonesia; dina.trisnawati@fp.umy.ac.id; 2Department of Plant Protection, Faculty of Agriculture, IPB University, Jl. Kamper, Kampus IPB Dramaga, Bogor 16683, Indonesia; fajrinfahmi@apps.ipb.ac.id; 3Center for Transdisciplinary and Sustainability Sciences, IPB University, Jl. Raya Pajajaran, Bogor 16153, Indonesia

**Keywords:** fall armyworm, integrated pest management, invasive pest, *Spodoptera frugiperda*

## Abstract

**Simple Summary:**

*Spodoptera frugiperda* is an invasive pest of corn in several countries, including Indonesia. The pest is highly polyphagous, with limited information available on its biological parameters. This information about its biological parameters, such as survival, life cycle, and fecundity using different host plants, could be critical to developing effective pest management strategies. After testing 14 different host plants, we found that even though *S. frugiperda* preferred corn as the primary host, several other host plants, e.g., papaya, water spinach, banana, spinach, cucumber, and a weed (coco grass), can act as alternate hosts for *S. frugiperda*, implying the use of these plants as a hedge or trap plant for the management of this pest using culture technique manipulation. However, other control methods need to be considered for further research to develop sustainable solutions for its management.

**Abstract:**

*Spodoptera frugiperda* is Indonesia’s relatively new invasive polyphagous insect pest. So far, *S. frugiperda* infestation has only been reported in corn and rice in Indonesia. However, *S. frugiperda* is known to feed on many commercial crops in other countries. To date, information on the biological parameters of *S. frugiperda* is limited in Indonesian ecologies. Since host plants are a critical factor for insect life history and have the potential to be used for pest control strategies, it is essential to study the biology and survival of *S. frugiperda* on different host plants. This research aimed to investigate the survival, development time, and fecundity of *S. frugiperda* on other host plant species to predict possible infestation rates for use in pest management. The study was conducted by rearing *S. frugiperda* on 14 common cultivated host plant species in Indonesia. The survival rate, development time, fecundity, and potential attack rate of *S. frugiperda* on various tested host plants were analyzed in this study. The findings revealed that corn was the primary host for *S. frugiperda*. The ability of *S. frugiperda* to survive on papaya, water spinach, banana, spinach, cucumber, and coco grass indicates that these plants are potential alternate hosts for *S. frugiperda*. Long beans, bok choy, choy sum, and beans might be indicated as a shelter for *S. frugiperda*. Meanwhile, inappropriate hosts for *S. frugiperda* include cabbage, broccoli, and cauliflower due to their low survival rate on these plants. This research indicates that these plants have the potential to be used as a hedge, trap, or bunker plant in *S. frugiperda* management strategies. However, to prevent detrimental damage, control methods are needed in an integrated manner, including monitoring pest populations, habitat manipulation, and conservation of natural enemies.

## 1. Introduction

Corn is the second staple food after rice for Indonesians [1]. The existence of pests, diseases, and cyclical climatic change are some of the challenges that farmers encounter when growing corn in Indonesia. According to Nurmaisah and Purwati [2], the Asian corn borer *Ostrinia furnacalis* Guenée (Lepidoptera: Crambidae), the corn cob borer *Helicoverpa armigera* Hübner (Lepidoptera: Noctuidae), the armyworm *Spodoptera litura* Fabricius (Lepidoptera: Noctuidae), the corn aphid *Rhopalosiphum maidis* Fitch (Hemiptera: Aphididae), and the shoot fly *Atherigona exigua* Stein (Diptera: Muscidae) are the major pests of corn in Indonesia. In the meantime, *S. frugiperda* has been noted as a new pest of corn in Indonesia since early 2019 [3].

America is where the fall armyworm (*S. frugiperda*) first appeared [4]. *Spodotera frugiperda* aggressively spread to the African Continent in 2016 [5] and Asian Continent in 2018 [6]. *Spodoptera frugiperda* was first discovered in the Asian Continent in India and spread to other Asian countries, including China, Taiwan, Japan, Cambodia, Malaysia, and the Philippines [5,6,7,8]. *Spodoptera frugiperda* was initially discovered in West Pasaman, Sumatra, Indonesia [3]. Currently, *S. frugiperda* has expanded across Indonesia’s entire islands [9].

*Spodoptera frugiperda* attacks corn as its primary host and could harm corn plants at any stage, including both the vegetative and generative stages [10]. At the same time, the most severe damage occurs in the vegetative phase [11]. *Spodoptera frugiperda* can attack the growing part of corn plants, causing the formation of young leaves or shoots to fail. Larvae that damage cobs can cause losses, reducing crop yields [12]. Despite having corn as its primary host plant, *S. frugiperda* has been reported to have a wide range of host plants, including 353 host plants from 76 plant families, such as Poaceae, Fabaceae, Solanaceae, Amaranthaceae, Brassicaceae, Caricaceae, Cyperaceae, Euphorbiaceae, and Cucurbitaceae [13]. Recently, *S. frugiperda* has also been reported to be infesting rice in several countries, including Indonesia [14]. However, facts in the field report that *S. frugiperda* more often feeds corn plants with different infestation rates [11,15,16]. Hill [17] mentioned that any pest has the potential to become a major pest, and every plant from a different region has its unique traits. Many species that cause significant damage to one crop may only have minor effects on other crops or even the same crop in a different region of the world.

Indonesia is an agricultural country with various landscape structures. Even though most paddy fields are used for growing rice [18], some farmers typically cultivate other crops such as corn, chilies, onions, beans, and green vegetables, in which the fields are surrounded by coconut, mango, cassava, bananas, papaya, or shrubs [19]. *Spodoptera frugiperda* has a wide variety of hosts, giving it the opportunity to attack various plants. Additionally, the adults of *S. frugiperda* can migrate from one location to another due to excellent flying ability [20]. However, whether all these host plants can support the development of *S. frugiperda* has yet to be known. Research on the life cycle of *S. frugiperda* in several host plants has been carried out in Indonesia [21,22,23,24,25] and other countries [20,26,27,28,29]. Research on the life cycle is critical because, in addition to the impact of plant species [26,30], geographic location can also affect insect bioecology differently due to varying climatic conditions [31,32]. Moreover, those studies only discussed the effect of diet (host plant) on the survival and development of *S. frugiperda*. More information is required to understand the survival, life cycle, and fecundity of *S. frugiperda* on different host plants. Do alternative host plants able to adequately support their growth? We hypothesized that some host plants of *S. frugiperda* only serve as a shelter, and some can be alternate hosts for *S. frugiperda*. An insect’s survival, life cycles, and fecundity will vary depending on the host plant. Therefore, we study the survival, development time, and fecundity of *S. frugiperda* using 14 potential host plant species and predict possible infestation rates for use in pest management.

## 2. Materials and Methods

### 2.1. Spodoptera frugiperda

*Spodoptera frugiperda* speciments used in this study were collected from corn fields around Kasihan, Bantul, Special Region of Yogyakarta, Indonesia. *Spodoptera frugiperda* was then reared in the laboratory insect rearing room (25 ± 1 °C, 80 ± 10% RH, 16L:8D) of the Department of Agrotechnology, Faculty of Agriculture, Universitas Muhammadiyah Yogyakarta. Larvae were kept solitary using a cup (6.4 in height × 3.8 cm in diameter) to avoid cannibalism. Larvae were fed with baby corn purchased at the market and cleaned using tap water. After the larvae transformed into a pupa, they were then transferred into a cylindrical adult cage (30 in height × 10 cm in diameter) covered with paper as an oviposition medium. Adult were reared until laying eggs and given a 20% honey solution on moistened cotton as a diet source. The eggs obtained were used for research purposes.

### 2.2. Diet Source

Fourteen plants were selected to be tested as a diet source for *S. frugiperda* (Table 1). These plants were selected based on the *S. frugiperda* host range database [20] and the variety of vegetables cultivated in Indonesia. Corn was used as a comparative host (control). Meanwhile, coco grass was used as a diet source based on preliminary observations of *S. frugiperda* on this plant during field observations. The plants that were used were obtained from the market or the field. Plants were given in the same form and growth stage. The plants were cleaned with tap water, dried, then cut into small pieces before being supplied as a dietary source.

### 2.3. Effect of Different Diets on the Survival, Development, and Fecundity of S. frugiperda

The selected host plants (diet) were applied as treatments. Each host plant treatment was replicated four times. One replication was tested using twenty larvae. In total, 56 experimental units and 1120 larvae were used during this research.

*Spodoptera frugiperda* eggs that had just hatched into first instar larvae were carefully transferred into a cup (6.4 in height × 3.8 cm in diameter) using a brush. Each replication (20 cups) was separated using a plastic tray. The prepared diet was given daily until the larvae reached the sixth instar. Pupae and adults were reared by the methods described above. The development of *S. frugiperda* was observed and recorded daily to determine each stage’s survival rate and development time, including eggs, larvae, pupae, and adults. When the insects became adults, the number of eggs laid (both viable and non-viable eggs) was counted daily until they died to measure fecundity.

### 2.4. Data Analysis

Kaplan–Meier survival analysis was used to assess the survival data [33]. The development time and fecundity data were subjected to stepwise simplification to determine the appropriate model based on the AIC value. The final model was analyzed using a general linear model (GLM) with the Gaussian family and log link function, except for pupal development time, diagnosed with the gamma family and identity function. Tukey’s HSD and multiple comparisons with Holm’s adjustment were used to further assess the mean difference between treatments [34]. Finally, the survival data of *S. frugiperda* on each plant was analyzed by multiple regression analysis using a correlation matrix to predict the possible infestation rate if the host plants tested were present simultaneously or intercropped in the field. R Statistics v 4.2.1 [35] was used for the statistical analysis, and the ggplot2 package was used to generate the graphs [36].

## 3. Results

The results showed that the host plant significantly affected the survival rate of *S. frugiperda* (*p* < 0.0001) (Figure 1). The highest survival rate (>85%) occurred when *S. frugiperda* was fed corn, papaya, and water spinach. In general, the survival rate decreased at the second and third instar larvae. It continued to decline until the adult stage, with an 88.75% final survival rate for *S. frugiperda* fed on corn and papaya and 86.25% for those on water spinach. The survival rate of *S. frugiperda* was also relatively high (80–85%) when they fed on banana (82.5%), spinach (81.25%), cucumber (80%), and coco grass (80%). *Spodoptera frugiperda* had a 71% survival rate when fed on bok choy and 77.5% on long beans. Conversely, the survival rate of *S. frugiperda* was only 47.5% and 51%, respectively, when fed on beans and choy sum. No *S. frugiperda* survived (0% survival rate) when fed on broccoli, cabbage, and cauliflower. Mortality began to occur when *S. frugiperda* at the fourth or fifth instar larvae when reared with cabbage and broccoli and at the pupal stage when fed on cauliflower.

The host plant also strongly influenced the development time of *S. frugiperda* (Figure 2). The development time of *S. frugiperda* from egg (GLM: F_10.33_ = 54.926, *p* < 0.001), larva (GLM: F_10.33_ = 46.837, *p* < 0.001), pupa (GLM: F_10.33_ = 23.741, *p* < 0.001), adult (GLM: F_10.33_ = 102.05, *p* < 0.001), as well as the total development time (F_10.33_ = 18.1, *p* < 0.001), varied significantly when they fed on a different diet. The development time was either longer or shorter compared to the development time in its primary host (corn). 

The eggs produced by adults, whose larvae fed on beans, coco grass, banana, spinach, bok choy, and choy sum, developed within two days, the same as those on the primary host (corn). Meanwhile, the eggs produced by adults whose larvae fed on cucumber, water spinach, papaya, and long beans, developed a day slower (three days). The development time of the larval stage shows exciting results. Larvae develop with the same development time when they are fed corn (18.87 days), papaya (17.90 days), and bok choy (18.50 days). The larvae developed two days faster when fed on choy sum (15.97 days) and long beans (16.27 days). Meanwhile, the larvae developed one day later when fed on spinach (19.17 days), two days on banana (20.98 days), three days on water spinach (21.92 days), nine days on cucumber (27 days), and twelve days on beans (31.32 days). The pupae developed within nine days when the larvae fed on corn. Pupae had a consistent development time when they fed on beans, coco grass, long beans, bok choy, papaya, and spinach. Pupae developed four days later when the larvae were fed on water spinach (13.27 days), five days on cucumber (14.64 days), and seven days on banana (15.85 days). Adults also lived for nine days when their larvae were fed on corn. Adults lived one day less when the larvae fed on bok choy (8.43 days), two days on choy sum (7.9 days), and three days on banana (6.42 days), beans (6.78 days), spinach (6.76 days), and water spinach (7.13 days).

However, adults raised from the larvae fed on long beans could survive one day longer (10.35 days), two days longer with cucumber (11.20 days), and up to five days longer with papaya (14.38 days) and coco grass (14.95 days). Overally, *S. frugiperda* developed for 37.58 days when they fed on corn. The total development time of *S. frugiperda* was two days shorter when they fed on spinach (35.12 days), three days on long beans (34.66 days), five days on bok choy (32.87 days), and even up to 12 days on choy sum (25.30 days). In contrast, the total development time of *S. frugiperda* was longer by almost four days when the larvae fed on banana (40.58 days), beans (41.72 days), and papaya (42.01 days), five days on coco grass (42.53), and six days on water spinach (43.18). Surprisingly, the total development time can take up to 13 days longer when the larvae are fed on cucumber (50.67 days).

There was also a significant effect of the host plant on the fecundity of adult females (GLM: F_10.33_ = 50.884, *p* < 0.001) (Figure 3). During her lifetime, a single adult female raised from corn-fed larvae could lay an average of 228,58 eggs. This number is not significantly different from those treated with spinach and bok choy. Adult females raised from papaya-fed larvae produced 51.45% more eggs (346.19 eggs) than those raised from corn-fed larvae. Moreover, females from the choy sum treatment also had 79.74% more eggs (410.85 eggs). Despite the longer adult life span of *S. frugiperda* fed on banana, beans, cucumber, coco grass, and water spinach, the fecundity in these treatments surprisingly showed opposite results. Adult females raised from cucumber-fed larvae could only produce 18.31 eggs. Meanwhile, adult females raised from coco grass-fed larvae could produce up to 609.87 eggs.

Multiple regression analysis shows different predictions of *S. frugiperda* infestation rate in the field when each plant is present together or planted in an intercropping system. For example, when broccoli and choy sum (r = 0.53) are intercropped, a moderate infestation rate may occur, as well as the correlation between one plant and another with a 0.5–0.6 correlation coefficient value. Meanwhile, other correlations coefficient of 0.7–0.8 indicate signs of a severe infestation rate and a very severe rate for those correlation coefficients value above 0.8 (Figure 4).

## 4. Discussion

In this research, we study the effect of different host plant species on the survival, development time, and fecundity of *S. frugiperda* using 14 potential host plant species to predict possible infestation rates for use in pest management.

The host plants had a substantial impact on *S. frugiperda*’s survival, development time, and fecundity. Although most of the plants examined in this study were found to be among the 353 species of host plants for *S. frugiperda*, only 11 out of the 14 plants evaluated in this study were able to support the development of *S. frugiperda* entirely from the egg to the adult. *Spodoptera frugiperda* was able to survive better when the larvae fed on corn as its host. However, several studies reported a lower survival rate of *S. frugiperda* fed on corn than that reported in this study [37,38,39,40]. Other investigations also reported a lower survival rate of *S. frugiperda* fed on beans [41] than that reported in this study. Possible causes of this result include plant parts used, varieties, and environmental conditions such as topography, temperature, and humidity [37,42]. The larvae fed on papaya had the same survival ability as corn. However, the leaves and seeds of papaya are reported to be effective insecticides for fall armyworms [43,44,45]. However, fall armyworms fed on papaya might lay more eggs than those fed on corn. Meanwhile, even though *S. frugiperda* had a high survival rate when fed on water spinach, *S. frugiperda* developed more slowly and had a lower fecundity than when fed on corn. Putra and Martina [25] also noted that *S. frugiperda*’s fecundity was significantly lower when fed on water spinach compared to when fed on corn.

The host plants tested in this study have been shown to influence the survival, development time, and fecundity of *S. frugiperda*. However, the results show that there is no relationship between the total development time and fecundity. For instance, the development time and fecundity of *S. frugiperda* fed on corn and spinach were comparable in this study. However, *S. frugiperda* fed on corn had a higher fecundity than *S. frugiperda* fed on spinach. In contrast, Maruthadurai and Ramesh [5] reported that the development time of *S. frugiperda* fed on spinach was longer than that fed on corn in India. While the fecundity of *S. frugiperda* fed on corn was also higher than that fed on spinach. The development time of *S. frugiperda* was longer, with lower fecundity when they were fed on water spinach, beans, and cucumbers. This might occur because *S. frugiperda* larvae that feed on water spinach, beans, and cucumbers develop more slowly than those fed on corn. In contrast, although the total development time of *S. frugiperda* that fed on papaya and coco grass was longer than that of *S. frugiperda* fed on corn, the fecundity of *S. frugiperda* on these plants was higher than that of corn because the development time of *S. frugiperda* larvae that feed on papaya and coco grass is relatively similar to that of *S. frugiperda* fed on corn, meaning that the nutrients obtained by the larvae are sufficient to support the growth and development of adults, which is indicated by the high fecundity. The shorter the insect life cycle, especially the larval phase, the better the nutritional quality of the food. This quality largely determines the initial step from larvae to adults. Thus, what the larvae eat at the beginning of their growth will significantly impact the fecundity of the female [37,38,42].

Other host plants, such as broccoli, cabbage, and cauliflower, were unable to fully support *S. frugiperda*’s development. The life cycle of *S. frugiperda* fed on broccoli, cabbage, and cauliflower stopped when the larvae were at the fourth and fifth instar. Thus, no *S. frugiperda* grown with those treatments developed to adults. In contrast, Wang et al. [46] reported that *S. frugiperda* could survive when reared on Chinese cabbage [*Brassica pekinensis* (Lour.) Rupr. var. Qinza 2], even though the survival rate was low. Wijerathna [41] reported that *S. frugiperda* could survive when reared on cabbage, even with low adult fecundity, which contrasts with those raised on bok choy (*Brassica rapa* subsp. *chinensis*) and choy sum (*Brassica rapa* subsp. *chinensis* var. *parachinensis*). Despite being of the same genus (*Brassica*), bok choy and choy sum resulted in a higher survival rate of *S. frugiperda* compared to those three plants. In addition, *S. frugiperda* also had a faster development time and more fecundity when fed on bok choy and choy sum compared to corn as its primary host. In addition, Putra and Khotimah [47] also noted that the life cycle of *S. frugiperda* was accelerated, and its fecundity increased when fed on bok choy, even in circumstances when corn was available as its primary host.

Almost all the host plants investigated in this study have commercial value except for coco grass, although this grass can be used as a fodder crop in Indonesia. Apart from this weed, other weeds, such as Napier grass, natal grass in Taiwan [26] and Guinea, and para grass in India [5], have also been reported to support the development of *S. frugiperda*. Weeds are competitors for cultivated plants. However, the fact that *S. frugiperda* can survive on some weeds creates a problematic situation. The presence of weeds can be a competitor for cultivated plants because, apart from affecting soil and plant nutrition [48], research results show that the presence of weeds (coco grass) can also potentially increase the infestation rate of *S. frugiperda*. However, weeds can be alternate hosts by using them as trap or hedge plants. Our results indicate that these plants can be used to determine control strategies such as intercropping or push–pull systems. There are other factors to be assessed, considering that the simultaneous planting (intercropping or polyculture) of these plants is possible to increase the infestation rate of *S. frugiperda*. Simultaneous planting can create a niche in which *S. frugiperda* can thrive due to the presence of secondary hosts. *Spodoptera frugiperda* will survive on the secondary host when the primary host is absent or the growing season changes. When the primary host’s planting season comes, *S. frugiperda* can return to the primary host [49]. Furthermore, the polyphagous nature of *S. frugiperda* can support transfers between one host plant and another [13]. For example, *S. frugiperda* infestation occurs on cotton in Brazil, where cover crops such as millet may act as an agent that exacerbates the attack rate in the following growing season [50]. An alternative control method is by regulating the planting time. The regulation of planting time is needed to break the life cycle of *S. frugiperda* in the field. For the next step, we intend to investigate *S. frugiperda*’s oviposition and feeding preferences in the examined plants to provide more information for future control strategies, considering that the possibility of using of an alternate host as trap or hedge plants would depend rather on the selectivity of ovipositing females than on suitability for larval feeding and development [51]. Indeed, these two parameters are often positively correlated [52], although exceptions to this rule were also reported [53].

## 5. Conclusions

In conclusion, the best host plant for *S. frugiperda* in Indonesia is corn. The ability of *S. frugiperda* to survive on papaya, water spinach, banana, spinach, cucumber, and coco grass indicates that these plants potentially become alternate hosts for *S. frugiperda*. Long beans, bok choy, choy sum, and beans can be categorized as a shelter for *S. frugiperda*. Meanwhile, cabbage, broccoli, and cauliflower are the least suitable hosts for *S. frugiperda*. This research implies the use of suitable hosts such as papaya, water spinach, banana, spinach, cucumber, coco grass, long beans, bok choy, choy sum, and beans as a hedge plant or trap plant for managing *S. frugiperda* in the field. Nevertheless, the control strategy must be carried out in an integrated manner. In addition, other related studies must be carried out to obtain more information for future control strategies.

## Figures and Tables

**Figure 1 insects-14-00629-f001:**
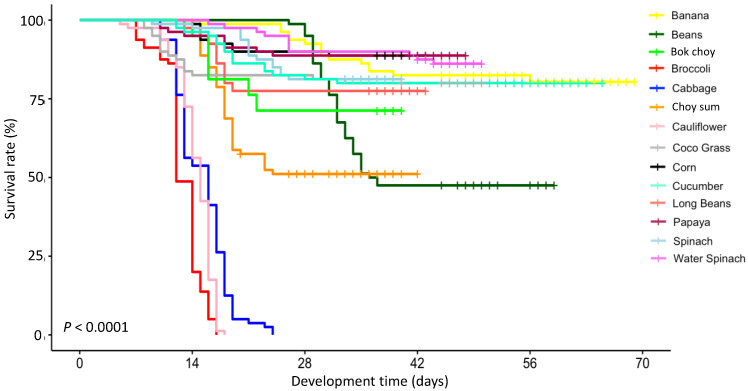
The survival rate of *Spodoptera frugiperda* fed on various host plant species.

**Figure 2 insects-14-00629-f002:**
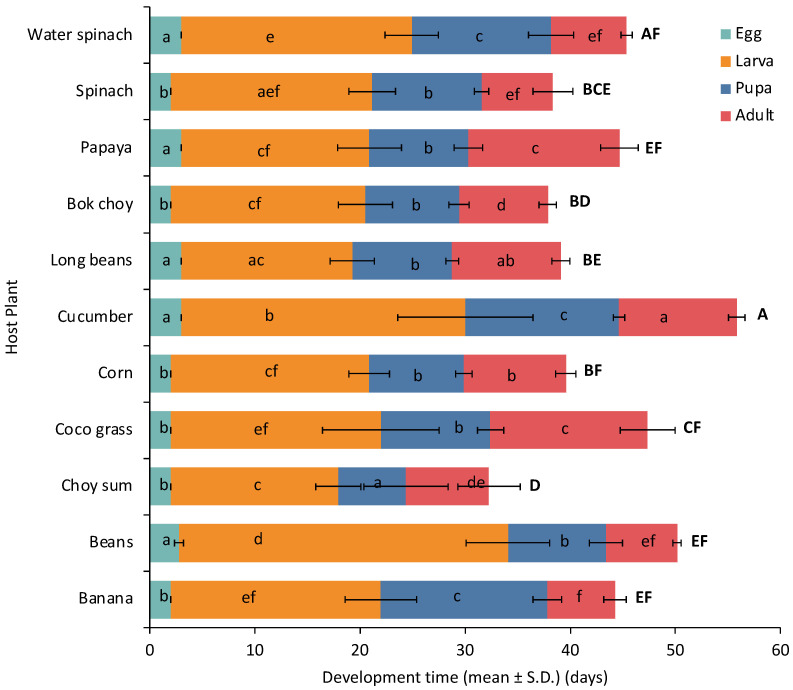
Development time of *Spodoptera frugiperda* on various host plant species. Means with different letters are significantly different by Tukey’s HSD test (α = 0.05). Lowercase letters indicate a difference in the same developmental stage between host plant treatments. Uppercase letters indicate a difference in total development time. SD, standard deviation.

**Figure 3 insects-14-00629-f003:**
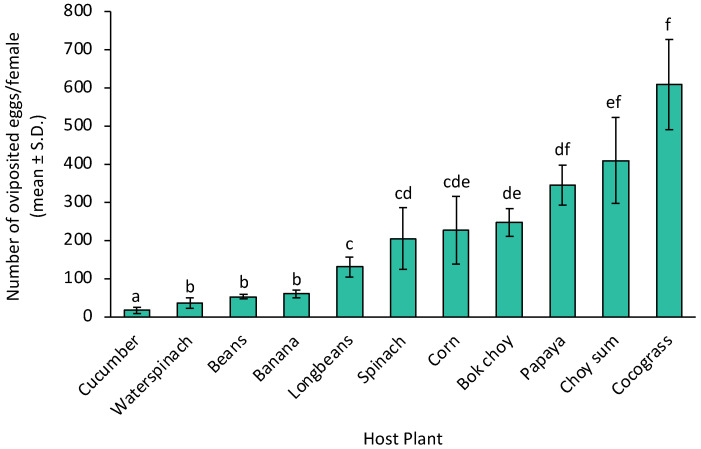
Fecundity of adult female *Spodoptera frugiperda* fed on various host plant species. Means with different letters are significantly different by Tukey’s HSD Test (α = 0.05). SD, standard deviation.

**Figure 4 insects-14-00629-f004:**
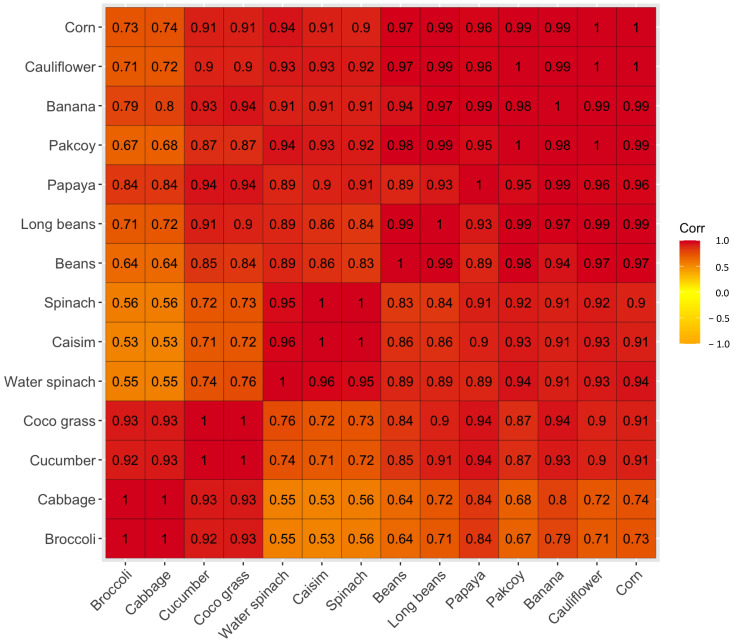
Heat map representation of possible infestation rates of *Spodoptera frugiperda* when the host plants are present simultaneously in the field or intercropped with each other. The numbers inside the squares are the correlation coefficients between two variables.

**Table 1 insects-14-00629-t001:** Host plants were used during the experiment.

**Host Plant**	**Family**	**Common Name**	**Diet Form**
*Zea mays*	Poaceae	Corn	Baby corn
*Cucumis sativus*	Cucurbitaceae	Cucumber	Leaf
*Brassica oleracea* var. *italica*	Brassicaceae	Broccoli	Bud
*Brassica oleracea* var. *botrytis*	Brassicaceae	Cauliflower	Bud
*Brassica oleracea* var. *capitata*	Brassicaceae	Cabbage	Bud
*Brassica rapa* subsp. *chinensis* var. *parachinensis*	Brassicaceae	Choy sum	Leaf
*Brassica rapa* subsp. *chinensis*	Brassicaceae	Bok choy	Leaf
*Phaseolus vulgaris*	Fabaceae	Bean	Pod
*Vigna unguiculata* ssp. *sesquipedalis*	Fabaceae	Long bean	Pod
*Amaranthus viridis*	Amaranthaceae	Spinach/ green amaranth	Leaf
*Ipomoea aquatica*	Convolvulaceae	Water spinach	Leaf
*Musa* sp.	Musaceae	Banana	Leaf
*Carica papaya*	Caricaeae	Papaya	Leaf
*Cyperus rotundus*	Cyperaceae	Coco grass	Leaf

## Data Availability

The data used in this study are available at https://doi.org/10.5281/zenodo.7978460 (accessed on 28 May 2023).

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
