# Peer review of "Survival, Development, and Fecundity of *Spodoptera frugiperda* (J.E. Smith) (Lepidoptera: Noctuidae) on Various Host Plant Species and Their Implication for Pest Management"

_insects, 2023, doi:10.3390/insects14070629_

Round 1

Reviewer 1 Report

This manuscript gives an insight into how different or non-likely preferential hosts can affect the survival, development and fecundity of S.frugiperda. The study and methodology is very well done and using a wide range of host plant gives a broader picture how this insect can spread and how it can be managed. However, certain minor comments/questions need to be addressed to make this study impactful.

1. Line 91/92: Were the diet form same during the whole experiment, were leaves/buds/pods of same growth stage or age when used as diet for the insect as using mature or young parts of plants can show a significant variations in the development of insect.

2. Line 165: Fecundity tells about how many eggs were laid by a healthy adult female insect they can be both viable and non-viable. Do the count of eggs in this assay are of viable eggs or a mix of both.

Do check the grammar for certain sentences for the manuscript . 

Author Response

Dear Reviewer,

We thank you for the helpful comments, advice, and corrections of this manuscript. We accept all those inputs and implement them in this revised manuscript. We have carefully considered each comment in track changes mode as requested by the Editor. Please select track change with no markup option to make it easier when you check the revised manuscript. We have made revisions in response to them, as follows. 

Comments and Suggestions for Authors

This manuscript gives an insight into how different or non-likely preferential hosts can affect the survival, development and fecundity of S.frugiperda. The study and methodology is very well done and using a wide range of host plant gives a broader picture how this insect can spread and how it can be managed. However, certain minor comments/questions need to be addressed to make this study impactful.

  1. Line 91/92: Were the diet form same during the whole experiment, were leaves/buds/pods of same growth stage or age when used as diet for the insect as using mature or young parts of plants can show a significant variations in the development of insect.

Response. Yes. It was the same. We used a similar part of a plant with the same growth stage to feed the larvae as revised in lines 136-137

  1. Line 165: Fecundity tells about how many eggs were laid by a healthy adult female insect they can be both viable and non-viable. Do the count of eggs in this assay are of viable eggs or a mix of both.

Response. We do not measure the hatchability rate of eggs. Therefore, we calculate both viable and non-viable eggs in this experiment as revised in line 154

Comments on the Quality of English Language

Do check the grammar for certain sentences for the manuscript . 

Response. The English in this final revised version has been checked by a colleague fluent in English. For a certificate, please see:  https://tinyurl.com/3cspa44z

Hopefully, our manuscript is now improved and acceptable to be published in Insects.

Thank you.

Yours sincerely,

Ihsan Nurkomar

Reviewer 2 Report

The research study aims to investigate the survival, life cycle, and fecundity of the invasive insect pest, Spodoptera frugiperda, on different host plants in Indonesia. The study is important as host plants play a critical role in the insect's life history and can be used in pest control strategies. The researchers reared S. frugiperda on 14 common cultivated host plant species in Indonesia and analyzed various parameters such as survival rate, development time, fecundity, and potential attack rate. Meanwhile, I have some concerns related to language and ways of data presentation and results interpretation (marked on the pdf file). 

The English language is poor for the manuscript and needs to be worked on for better presentation of results! 

Author Response

Dear Reviewer,

We thank you for the helpful comments, advice, and corrections of this manuscript. We accept all those inputs and implement them in this revised manuscript. We have carefully considered each comment in track changes mode as requested by the Editor. Please select track change with no markup option to make it easier when you check the revised manuscript. We have made revisions in response to them, as follows.  

Comments and Suggestions for Authors

The research study aims to investigate the survival, life cycle, and fecundity of the invasive insect pest, Spodoptera frugiperda, on different host plants in Indonesia. The study is important as host plants play a critical role in the insect's life history and can be used in pest control strategies. The researchers reared S. frugiperda on 14 common cultivated host plant species in Indonesia and analyzed various parameters such as survival rate, development time, fecundity, and potential attack rate. Meanwhile, I have some concerns related to language and ways of data presentation and results interpretation (marked on the pdf file).

Line 18. Change several host plant to several other host plants

Revised in line 20

Line 25. Change attack to feed on

Revised in line 28

Line 28-29. Re-write

Revised in line 33-35

Line 49-50. Re-write

Revised in line 69-70

Line 54. Change occurred to occurs

Revised in line 74

Line 55. Change points to phase or part

Revised in line 75

Line 74-77. Re-write

Revised in line 101-113

Line 82. Change rh to RH

Revised in line 119

Line 86. Change transfer to transferred

Revised in line 124

Line 91. Purposively?

Removed

Line 94. Change also to further

Revised in line 134

Line 95-96. Re-write

Revised in lines 135-138

Line 100. Replicate or replicate group ?

Revised in lines 142-144

Line 105-106. This information if mentioned before and getting repeated, make sure to limit this kind of limitations.

Revised in line 150

Line 123. Change attack to feeding. Intercropped?

We revise the attack to infestation rate. The regression analysis was performed to predict the infestation rate when each plant is present simultaneously or intercropped as revised in lines 168-171

RESULT. Authors need to present data more scientifically. The mean values with standard deviation/error make better sense.

All mean values are included in the text (lines 193-221) and graph. The standard deviation is already included in the graph.

Line 146. Change slower to shorter.

Revised in line 199

Line 151-154. Re-write

Revised in lines 204-211

Line 159-164. Results in general are presented more vaguely. They needs to be presented in more scientific manner with Standard deviation or error values, if possible.

All mean values are included in the text (lines 193-221) and graph. The standard deviation is already included in the graph.

Line 166. Re-write

Revised in lines 242-244

Line 168-173. Any possible reason why fecundity and survivability was not correlated!

As described in lines 198-322

Line 178-180. Not clear enough!

Revised in lines 258-266

DISCUSSION. In general, the discussion if quite superficial and mere representation of data. What's lacking to me is the deep understanding of results and its ecological importance. Authors need to think how the results make more sense in terms of S. frugiperda management and control strategies.

Line 189. “attack” seems very inappropriate word so needs to be changed into entire manuscript.

Revised in line 84

Line 192-193. Fragmented sentences, not clear enough!

Removed

Line 195-197. Very vague and needs to be rewritten! Further I feel this sentence more belongs to introduction rather discussion.

Revised in lines 107-109

Line 217. offspring is better choice than fecundity

Not agree. Offspring comes from viable eggs. Meanwhile, we count both viable and non-viable eggs in this study as described in line 154.

Line 218. Authors need to include those studies/citations.

Revised in lines 285-287

Line 225-231. Any possible explanation for the discrepancy of results?

Yes. As we revised in lines 298-322

Line 235-238. The sentence needs to be reworded and corrected for language!

Revised in lines 324-329

Line 263. very confusing !

Revised in lines 356-359

Line 276-279. These two sentences are very contradictory to each other and needs to be worked on.

Removed

Hopefully, our manuscript is now improved and acceptable to be published in Insects.

Thank you.

Yours sincerely,

Ihsan Nurkomar

Reviewer 3 Report

The aim of the present study was to estimate several main biological parameters (survival, rate of development, fecundity) of the invasive polyphagous moth Spodoptera frugiperda with feeding on different acceptable host plants. With this aim, the authors have conducted a series of simple experiments and obtained rather clear results which can be important for the elaboration of the methods for control of this pest. The experiments were well planned and conducted, the data were properly analyzed. Thus, the manuscript can be published although it still needs some minor corrections (see below).

Table 1: Please, indicate not only species but also variety of used host plants, because even polyphagous insects often prefer some varieties and reject other varieties of the same cultural plant species. If possible, it would be also useful to indicate the supplier (for bought seeds) or locality (for field-collected plants). As cited by the authors (line 190) “every plant from a different region has its own unique traits”. Of course, varieties are even more important (line 214).

Line 96: What is the meaning of “Plants were prepared as above..“. Does it mean cleaned with water (lines 85-86). Anyway, this should be described as well as other details of the methods such as were any water sources (as wet cotton) given to the leaves, etc.

Line 146: I am also not a native speaker, but I think that “longer time” means the same as “slower time”. Therefore, in this case, the phrasing “either ..... or” can’t be used.

Lines 183-206: This paragraph is not the Discussion of the results obtained by the authors, but rather the explanation of the reasons of their study. In fact, the Discussion of the results started from line 207. It is not mandatory, but, possibly, lines 183-206 should be either deleted or replaced to Introduction?

Line 259: The possibility of the use of weeds (and other alternate hosts) as trap plants would depend rather on selectivity of ovipositing females than on suitability for larval feeding and development. Indeed, these two parameters are often positively correlated, but exceptions to this rule were also reported.

Figure 2: It would be also interesting to the readers to somehow indicate statistical significance of pairwise difference in total time of pre-adult development (not only in egg, larval, and pupal stages).

Figure 3: It is not very important, but the order of letters (from low to high fecundity) is a bit strange: a, f, b, c ...

Figure 4 is not mentioned in the text (or at least I can’t find it). However, just this figure is not very clear (at least for me) and definitely needs explanations.

Author Response

Dear Reviewer,

We thank you for the helpful comments, advice, and corrections of this manuscript. We accept all those inputs and implement them in this revised manuscript. We have carefully considered each comment in track changes mode as requested by the Editor. Please select track change with no markup option to make it easier when you check the revised manuscript. We have made revisions in response to them, as follows. 

Comments and Suggestions for Authors

The aim of the present study was to estimate several main biological parameters (survival, rate of development, fecundity) of the invasive polyphagous moth Spodoptera frugiperda with feeding on different acceptable host plants. With this aim, the authors have conducted a series of simple experiments and obtained rather clear results which can be important for the elaboration of the methods for control of this pest. The experiments were well planned and conducted, the data were properly analyzed. Thus, the manuscript can be published although it still needs some minor corrections (see below).

Table 1: Please, indicate not only species but also variety of used host plants, because even polyphagous insects often prefer some varieties and reject other varieties of the same cultural plant species. If possible, it would be also useful to indicate the supplier (for bought seeds) or locality (for field-collected plants). As cited by the authors (line 190) “every plant from a different region has its own unique traits”. Of course, varieties are even more important (line 214).

Response. Unfortunately, we did not recognize the variety of plants used. However, we used the same form and growth stage of plant as described in lines 136-170

Line 96: What is the meaning of “Plants were prepared as above..“. Does it mean cleaned with water (lines 85-86). Anyway, this should be described as well as other details of the methods such as were any water sources (as wet cotton) given to the leaves, etc.

Revised in lines 137-138

Line 146: I am also not a native speaker, but I think that “longer time” means the same as “slower time”. Therefore, in this case, the phrasing “either ..... or” can’t be used.

Response. We mean some larvae develop faster and others are slower. Therefore, we used that phrasing. We change slower to shorter to make it clearer as revised in line 199

Lines 183-206: This paragraph is not the Discussion of the results obtained by the authors, but rather the explanation of the reasons of their study. In fact, the Discussion of the results started from line 207. It is not mandatory, but, possibly, lines 183-206 should be either deleted or replaced to Introduction?

Removed to introduction in lines 101-113

Line 259: The possibility of the use of weeds (and other alternate hosts) as trap plants would depend rather on selectivity of ovipositing females than on suitability for larval feeding and development. Indeed, these two parameters are often positively correlated, but exceptions to this rule were also reported.

Revised in lines 367-374

Figure 2: It would be also interesting to the readers to somehow indicate statistical significance of pairwise difference in total time of pre-adult development (not only in egg, larval, and pupal stages).

Revised

Figure 4 is not mentioned in the text (or at least I can’t find it). However, just this figure is not very clear (at least for me) and definitely needs explanations.

Revised in line 265

Comments on the Quality of English Language

The English language is poor for the manuscript and needs to be worked on for better presentation of results!

Response. The English in this final revised version has been checked by a colleague fluent in English. For a certificate, please see:  https://tinyurl.com/3cspa44z

Hopefully, our manuscript is now improved and acceptable to be published in Insects.

Thank you.

Yours sincerely,

Ihsan Nurkomar

Round 2

Reviewer 2 Report

Improved! Although I have some minor suggestion to make for figure (marked on the file). 

None

Author Response

Dear Reviewer,

We thank you for the helpful comments, advice, and corrections of this manuscript. We accept all those inputs and implement them in this revised manuscript. We have carefully considered each comment in track changes mode as requested by the Editor. Please select track change with no markup option to make it easier when you check the revised manuscript. We have made revisions in response to them, as follows.  

Line 18. I will prefer keeping it world-wide. Why to limit it only to Indonesia?

Respond. We limit this to Indonesia only because each region has a different agricultural landscape structure as described in Introduction. However, we agree to revise it as Indonesia’s landscape looks like other countries, especially in Southeast Asia. So, we delete it.

Line 255. The letters marking post-hoc analysis are not in the correct order.  Waterspinach, beans and Banana should have been marked as "b" rather than "f" which way it could have been easy to understand. The whole idea is to make it in order so easy to grasp.

Respond. Revised

Hopefully, our manuscript is now improved and acceptable to be published in Insects.

Thank you.

Yours sincerely,

Ihsan Nurkomar